# A Comparison of 5v5 and 3x3 Men’s Basketball Regarding Shot Selection and Efficiency

**DOI:** 10.3390/ijerph192215137

**Published:** 2022-11-16

**Authors:** Zoltan Boros, Kata Toth, Gergely Csurilla, Tamas Sterbenz

**Affiliations:** 1School of Doctoral Studies, Hungarian University of Sports Science, 1123 Budapest, Hungary; 2Sport Economics and Decision Making Research Centre, Hungarian University of Sports Science, 1123 Budapest, Hungary; 3Institute of Economics, Centre for Economic and Regional Studies, 1097 Budapest, Hungary

**Keywords:** basketball, 3x3, behaviour, offensive efficiency, offensive rating, shot selection, decision-making

## Abstract

Background: Both in 5v5 and 3x3 basketball, the goal of the players is to score more points than the opponent. However, the differences in rules between two basketball disciplines can affect thinking, behaviour, and decisions of the players. A core difference between two disciplines is the value of the shots. In 5v5, long-range shots are worth three points and close-range two, while in 3x3, their values are two and one points, respectively. As the value ratio of the close and long-range shots is greater in 3x3, we assume that players make different decisions about their shot selection in 3x3 than in 5v5, which can affect offensive efficiency. Methods: We analysed game statistics of the 2019 men’s 5v5 and 3x3 Basketball World Cups. Besides regular statistical indicators, we applied relative offensive rating to be able to compare the two disciplines. Results: The analysis of relative offensive rating showed that offences are more effective in 5v5 than in 3x3. We also found significant difference in shot selection and efficiency. In 3x3, there is a higher proportion of the shots than in 5v5, but long-range shots are more successful in 5v5. Conclusions: For rule differences that affect player’s shot selection and affect offensive efficiency, their decisions are characterized by ecological dynamics and naturalistic decision-making.

## 1. Introduction

Three on three (3x3) basketball officially debuted at the Singapore Youth Olympics in 2010 (www.fiba.basketball). This new sport became immediately widespread over the world and gained immense popularity, and 3x3 has already become part of the Olympic program in Tokyo 2020.

Some previous studies [1,2,3,4,5,6] had already found a significant difference in physical intensity and shooting efficiency between 3x3 and traditional basketball (5v5), but the comparison has not been made in terms of players’ thinking, decision making, motivation, shot selection, ball possession, and offensive efficiency.

In terms of strategic, tactical, and technical elements, the basics of 3x3 are based on 5v5 basketball, but there are differences in rules (FIBA) [7]. The 3x3 and 5v5 disciplines differ in court size, number of players, game time, and shot clock, which is why this comparison has increased importance. It is important to note, on the one hand, areas of shots are the same in the two basketball disciplines, and, on the other hand, all 3x3 players come from traditional 5v5 basketball. The value of shots taken from within (close shot) and outside (long-range shot) the arc (5v5: three and two points, respectively; 3x3: two and one points, respectively) and the ratio of values are different in 5v5 (3:2) and 3x3 (2:1) (FIBA) (Figure 1). Another core difference is that the shot clock differs between the two disciplines. It is 24 seconds in 5v5, while it is 12 seconds in 3x3.

Due to the 3x3 rules, the attacker–defender role can change continuously within a few seconds, which requires players to maintain constant concentration. The immediate role changing can change the behaviour and thinking of the players in 3x3 as it does in 5v5. All of these suggest players’ decision making is different in a 3x3 game than in 5v5, which may be interesting for this kind of comparison of the two forms of basketball.

In terms of shot selection and effectiveness, previous studies [3,4] found differences in comparing 3x3 and 5v5, which found that the share of long-range shots was higher and less effective in 3x3 than in 5v5. However, they did not research the reason for what makes players select more long-range shots, even though these are less effective.

Basketball is often considered to have an unpredictable environment [8], as are team ball sports, due to its variation in time and complexity [9], which requires players to respond effectively to uncertain situations. In team ball games, players need to constantly adjust their decisions and actions to the behaviour of others to achieve success in a competitive environment [10]. They must consider their opponent, teammates, coaching instructions, interpersonal distances, and court lines. In a rugby union study [11], researchers demonstrated that affordances are often perceived during the unfolding action in a game. For example, the movement of a teammate or opponent can create a new opportunity for another action during the game that overrides the pre-planned action.

To understand the decision mechanisms, three perspectives have emerged information processing, ecological dynamics, and naturalistic decision-making. Information processing is determined by cognitive abilities during the decision-making process and revolves around player’s access to memory representations. From information-processing stores, players make decisions through the process of selecting formalized responses [12]. According to the ecological-dynamics perspective, players make decisions through perception, where perception and action are coupled through information available in the environment, arising during continuous interactions between performer and environment [13]. The behaviour of the players is influenced by the continuous contact within the team and adaptation to the opponent [14,15,16,17], which limits decision-making, so decisions are determined in relation to the structure of the environment and the executive’s abilities. Decisions made through experience are characterized by the naturalistic decision-making perspective [18,19], where human performance occurs under pressure, in time-limited situations, so decisions are made through a process based on recognition, which changes from situation to situation [18,19]. 

These approaches show that the decision-making process depends on factors such as complexity, typicality, available time, and available contextual priorities in the game situation [11].

### 1.1. Intensity

Research [1,2,5,6] connected to 3x3 basketball only started in last decade and mainly focused on players’ physical and physiological characteristics. Physical requirements of 5v5 and 3x3 basketball were compared with the help of GPS technology [5]. According to this research in 3x3 basketball, players showed significantly higher values of distance covered, speed, and load. Players covered greater distances and their movement interchanged between acceleration and deceleration more frequently per minute.

Another research study [6] found that player’s number of ball contacts was higher in 3x3 games, suggesting greater involvement of players in the game. 

The 3x3 games specifically require high-speed inertial movement within a limited distance, resulting in a relatively high physiological response [5], so players must make decisions under such conditions. As a result, intuition appears in the form of the take-the-first heuristic [20,21,22], in which the identification of predictive signals, anticipation [20,21,23], and option generation play a role. In these situations, the speed of the information processing and decision mechanism are decisive, which validates the performance of elite players and differentiates them from less-skilled elite players [14,24,25]. Elite players have excellent skills; they extract and process signals from the environment [26] and recognize and interpret known game situations [27,28].

### 1.2. Performance-Indicator Game Statistics

The statistical system applied during basketball games, due to the nature of modern sports, fulfils the interest of media and spectators and helps in performance evaluation of both individual players and teams [29]. In sports, each game’s statistical system plays an important role in motivating players, which determines aspects of their behaviour; it also impacts how they think, feel [30], and evaluate their performance. In addition, game statistics serve as a basis for the decisions of sports leaders [29,30]. Basketball professionals put great emphasis on game statistics as a tool for evaluating player and team performance, which can provide theoretical explanation for the outcome of the game [30].

In addition to common game statistical indicators, like shooting efficiency, Oliver’s [31] ball-possession formula (Equation (1)) can be applied to determine the offensive rating (Equation (2)) to differentiate among teams, not only for a game but for an entire season or World competition as well. In 3x3 basketball, this formula has not been investigated before. Another goal of our study is to apply Oliver’s formula in 3x3 basketball and compare the two basketball sports in offensive rating.

Official game statistics record data mostly on offense; the most frequently used parameter is effective field goal percentage (eFG%) (Equation (1)) [32]:(1)$$\mathrm{eFG}\%=\frac{\mathrm{FG}+0.5\cdot 3P}{\mathrm{FGA}}$$
where FG is the made field goal, 3P is the made three-point value, and FGA is the total field-goal attempts.

According to Oliver [31], while stressing the importance of analysing individual and team game statistics, he warns that there are parts of the basketball game that are difficult to measure with game statistical indicators, such as defence. In basketball, to prepare for games, to build the best tactics, and to make good decisions during the game, coaches need to know which elements of the game are the most crucial [33,34,35,36]. By analysing offense and defence in unity, advanced sports analytical methods help coaches, as decision-makers, to develop rational strategies and tactics. This is especially important in case of close (tie) games, where there are small differences between the performance of the two teams. In close 5v5 games, differences were found between winning and losing teams in several game statistical performance indicators [35,37].

For analysing key game statistical performance indicators in 5v5, games can be classified into three categories (close, balanced, and unbalanced games) based on their outcome [35,37]. Winning and losing matches are distinguished by number of rebounds, ratio of steal balls and turnovers, and the two- and three-point shooting efficiencies [35,37]. An analysis [38] focusing only on close games revealed that winning teams attempted significantly fewer three-pointers and shot them with higher efficiency than losing teams. In addition, a greater number of defensive rebounds, a greater number of free-throw attempts, and higher free throws also contributed to success. A study [36] has identified the key game statistical indicators in 3x3 basketball. Winning teams were found to make fewer turnovers and allow fewer rebounds for the opponent; therefore, coaches need to focus on minimizing the opponent’s ball possessions [36,37].

In 5v5 game statistics, shooting efficiency, rebounds, turnovers, and other objective data are used to compare teams. Better performance can theoretically be explained by the offensive and defensive efficiency ratings, both of which are based on ball possession [38]. This method allows time-related variables to be ignored, making comparisons easier by means of offensive and defensive efficiency ratings. Offensive efficiency of a team is shown by its offensive rating (points scored per possession), while defensive efficiency is displayed by its defensive rating (points allowed per possession). As for the offensive value in the NBA between 1987–2009, the 22-year season average was 1.02 [39]. This increased to 1.08 in the 2019/2020 NBA season, so, on average, teams score approximately one point per ball possession. When relating the offensive rating (points per possession) of professional men’s 5v5 basketball to the maximum points that can be achieved during an attack, which is three points (disregarding extreme game situations), it can be stated that the offensive efficiency in 5v5 is currently approximately 35%. These indicators have not been examined in 3x3 so far, so one of our goals is to examine this value and compare it to 5v5.

### 1.3. Strategic and Tactical Decision-making

Ball possession can end in a basketball game in three ways: a shot, a lost ball (turnover), or a penalty shot. Considering a shot, players decide the value of the shot to be taken depending on the team’s strategy, tactics, and the given situation. A long-range shot has a higher value but lower rate of success, whilst a close shot is worth less but comes with greater probability of success.

In certain game situations, players may think differently at different times in the match. In contrast to the model of perfectly rational decisions, the phenomenon of loss aversion [38] has been shown during the statistical analysis of the NBA games. NBA players were found to take different risks depending on whether their team led or was at a disadvantage in the game, i.e., they had different attitudes in shooting situations. Unlike the model of rational decisions, risk-taking can also be influenced by the instantaneous result of the game, so a different decision-making perspective can characterize the players’ decisions on the court.

Based on the statistical data of the analysis of 5v5 basketball games, it can be clearly stated that due to the 1.5 ratio of two- and three-point shots, the number of attempts and successes of three-point shots have become a determining factor [40]. This is based on the change in the selection of shot locations in the NBA; over the past two decades, the number of mid-range shots drastically decreased, while more-effective close shots and higher-value long-range shots dominated [41].

Players prefer multi-point shooting whenever it is possible because, like all competitive athletes, they strive to maximize their own performance by optimizing everything [41], be it movement, decision-making, proper mental status, or any segment of their sport that can affect their success.

The player’s decisions with the ball are influenced by the number of times the player is in a decision-making position. Compared to 5v5, there is significantly more ball possession per player in 3x3 [6]. In 5v5, the ball-possession rate of the playmaker is the highest. However, in 3x3, the smaller number of players and offensive time (shot clock) reduce the role of the primary ball possessor. In 3x3, individual ball possession is more evenly distributed within the team than in 5v5, and due to fewer players, the differences between the role of posts are faded.

### 1.4. Aims

According to previous comparative-intensity and decision-making studies [1,2,3,4,5,6], both in 5v5 and 3x3, shooting attempts are made from the same distances (due to the same court lines), but shots are taken under different physiological influences. Therefore, the aim of this study is to compare the two types of basketball in terms of shot selection, shooting efficiency, ball possession, offensive rating, and relative offensive rating. 

During the game, three and two points can be scored per ball possession in 5v5, two and one points in 3x3, and the shot clock is 24 seconds in 5v5 and 12 seconds in 3x3. Based on these rule differences, we assumed that offensive efficiency and shot efficiency differ between the two basketball disciplines. Regarding the selection of shots, we assume that the larger point difference achieved in 3x3 (5v5: 3/2 = 1.5x; 3x3: 2/1 = 2x) affects the player’s decision mechanism. Furthermore, we assumed that due to the difference in the rules between the disciplines, the different ratio of points available per ball possession results in a difference in the ratio of close- and distant-shot attempts. Finally, we assumed the relative-offensive-value difference between 3x3 and 5v5, from which we can conclude the decision-making mechanism of the players, shows which aspects characterize them.

## 2. Materials and Methods

### 2.1. Data Collection

Data of the 2019 men’s Basketball World Cups (WC), both the 3x3 and 5v5 disciplines, were collected from the official basketball website FIBA and by 3x3 WC video analysis.

From the 3x3 WC, data of 20 teams in 48 games were collected, while from the 5v5 WC, data of 32 teams in 92 games were available. Teams in each discipline were coded separately; for instance, team USA had different codes in 3x3 and 5v5, appearing as different teams. Games were grouped based on the score difference [23] and the outcome, yielding six categories: close win, close loss, balanced win, balanced loss, unbalanced win, and unbalanced loss (Table 1).

### 2.2. Data Processing

The following variables were calculated from the raw dataset: possession, points per possession, relative points per possession, shooting percentages for both the close- and long-range shots, and the ratios of close- and long-range-shot attempts to all field-goal attempts. For 5v5 and 3x3 basketball, possession (P) (Equation (2)) was calculated with Oliver’s formula [31], which is used in NBA statistics:(2)P=FGA+TO−OR+0.436 x FTA
where FGA is the number of field goal attempts, TO is the turnovers, OR is the offensive rebounds, and FTA is the free-throw attempts. In addition, we also applied Oliver’s offensive-rating formula (Equation (3)) to both basketball disciplines: (3)$$\mathrm{Offensive}\;\mathrm{rating}=\frac{\mathrm{Points}\;\mathrm{scored}}{\mathrm{possesion}}$$

Finally, we compared the relative offensive rating (Equation (4)) of the two disciplines, which is the ratio of the points scored per ball possession to maximum points achievable per ball possession. It can be calculated as follows: (4)$$\mathrm{Relative}\;\mathrm{offensive}\;\mathrm{rating}=\frac{\mathrm{Point}/\mathrm{Possession}}{\mathrm{maximum}\;\mathrm{achievable}\;\mathrm{points}\;\mathrm{from}\;a\;\mathrm{possesion}\;}$$

In the case of 3x3 basketball, all games were manually analysed. Thus, in the previous equation, the exact number of possessions related to FTA was available. Points per possession and relative points per possession (Equation (4)) were further calculated.

Statistical analysis was carried out in the R open-source (R 4.1.2, R Core Team, Vienna, Austria) environment. Descriptive statistics are given in mean and standard error of mean (SEM) (Table 2). Differences are indicated with mean and 95% confidence interval of mean (95% CI). Data visualization was performed with a regular boxplot showing minimum, lower quartile, median, upper quartile, and maximum, while outliners were marked with dots. Mixed linear-regression models were constructed to analyse the research questions. In these models, depending on the research question, fixed factors were the discipline of basketball and range of the shots. Random effects were considered in all cases for team and game type, allowing for both random intercept and slope. Groups defined by the fixed factors were compared using planned comparisons with one-step *p*-value correction. The level of significance was set at α = 0.05 in all cases.

## 3. Results

### 3.1. Descriptive Data

The parameters used during the analysis are presented in Table 2.

**Table 2 ijerph-19-15137-t002:** Descriptive data of considered parameters (mean ± SEM).

	5v5 (*n* = 184)	3x3 (*n* = 96)
points scored	79.5 (1)	16.8 (0.4)
possession	76.6 (0.4)	30.3 (0.5)
offensive rating	1.04 (0.01)	0.55 (0.02)
relative offensive rating	34.7% (0.4%)	27.6% (0.8%)
overall shooting efficiency	44.5% (0.6%)	41.5% (1,1%)
close-shot efficiency	50.9% (0.7%)	54.2% (1.5%)
long-range-shot efficiency	33.8% (0.8%)	25.2% (1.5%)
close shots/overall shot attempts	62.3% (0.6%)	55.7% (1.2%)
long-range shots/overall shooting attempts	37.7% (0.6%)	44.3% (1.2%)

### 3.2. A Comparison of Relative Offensive Rating

To compare relative offensive ratings between the two disciplines, the mixed linear model was specified with fixed effect of the discipline. The model showed that discipline affected relative offensive rating (F(1, 5.1) = 36.8, *p* = 0.002). In the 5v5 WC, the relative offensive rating was greater than that of the 3x3 WC, with an average of 7.1% higher (95% CI: 4.8%–9.3%) (Figure 2).

### 3.3. A Comparison of Shooting Efficiency

Shooting efficiency was analysed across disciplines, ranges, and their combination. For that, the mixed linear model contained discipline, range, and their interaction as fixed effects. According to the model, shooting efficiency was affected by range (F(1, 510.1) = 619.3, *p* < 0.001) and the interaction of range and discipline (F(1, 510.1) = 41.6, *p* < 0.001) but not discipline alone (F(1, 13.2) = 3.3, *p* = 0.094). Pairwise comparison revealed that overall shooting efficiency did not differ between the disciplines (Figure 3A). However, shooting efficiency for close–mid-range shots was 21.4% greater than long-range shots (95% CI: 18.6%–24.1%, *p* < 0.001) (Figure 3B). In the case of discipline-range interactions, only the same ranges were contrasted between the disciplines. Close–mid-range shots did not show any difference (3x3: (54.2 ± 1.5)%, 5v5: (50.9 ± 0.7)%, *p* = 0.131) between the two disciplines, while long-range shots were 8.6% better in the 5v5 discipline than in the 3x3 one (95% CI: 4.5%–12.7%, *p* < 0.001) (Figure 3C).

### 3.4. A Comparison of Selection of Shots

Shots selection by means of ratio of close–mid- and long-range-shot attempts to all field attempts was modelled with a mixed linear model, where range and discipline-range interactions were the fixed effects. The percentage of all field-goal attempts was affected by range (F(1, 556) = 459.2, *p* < 0.001) and the interaction of range and discipline (F(1, 556) = 60.6, *p* < 0.001). The overall percentage of close–mid-range shots was (52% ± 0.7%) higher than long range shots, with an average of 21.3% higher (95% CI: 19.2%–23.3%, *p* < 0.001) (Figure 4A). In the case of the 5v5 discipline, the percentage of long-range-shot attempts from all field-goal attempts was lower than in 3x3, with an average of 6.5% lower (95% CI: 3.9%–9.2%, *p* < 0.001) (Figure 4B).

## 4. Discussion

The innovation of our study is the comparison of the two types of basketball (3x3 and 5v5) from a whole new perspective. One of the aims of our study is to reveal whether there is a difference between 5v5 and 3x3 in the players’ shot selection and efficiency. The other aim of our study is to investigate whether the different value ratios yield different proportions of close- and long-range-shot attempts. Through these results, the offensive efficiency and relative offensive value of the two disciplines was determined, which made the two basketball forms comparable in terms of offensive efficiency. Due to the differences in the rules, we assumed that the difference in the maximum point value per possession affects the players’ shot selection. We examined the areas of study from which the two basketball forms can be compared. The analysis of the variables revealed important game statistics regarding the players’ decision-making and the change in their behaviour when the game rules, conditions, and environment change.

For our research, we used basketball-game statistical indicators and examined ball possession and points scored, the ratio of relative offensive rating, shooting efficiency, shot selection, and long-range and close shots in the two disciplines.

### 4.1. Relative Offensive Rating

The maximum points that can be scored during an offense are three points in 5v5 (except in extreme cases, such as a free throw made for a foul committed after a long-range shot made), while it is two points in 3x3. In our research, we applied the offensive rating (used for the effectiveness of the offenses in the 5v5 games) for 3x3 games as well, but for the sake of better comparison, we used the relative offensive rating based on the different scoring in the two disciplines. In 5v5, the possession of the ball was approximately 77 (76.6 ± 0.4) per game, while the average score of a team was nearly 80 points (79.5 ± 1), which shows an offensive rating of 1.04 (± 0.01) and confirms the previously measured 1.02 offensive rating [33,41]. In 3x3, the ball possession was nearly 30 (30.3 ± 0.5) per game, of which the teams scored an average of 16.6 (± 0.4) points, resulting in a 0.55 (± 0.02) offensive rating. This shows that in 3x3 basketball, teams scored an average of one point per two ball possessions. Comparing the relative offensive ratings of the two disciplines, the 5v5 offensive efficiency index was 35% (34.7% ± 0.4%), which means that the teams scored approximately one point out of three points per ball possession. This rating was 27% (27.6% ± 0.8%) in 3x3, so it can be stated that 5v5 has a significantly higher efficiency index than 3x3.

### 4.2. Shooting Efficiency

In terms of the effectiveness of all shots, there was no significant difference between the two basketball sports (5v5: 44.5% ± 0.6%; 3x3: 41.5% ± 1.1%). In 3x3, the efficiency of close shots is better than in 5v5 (5v5: 50.9% ± 0.7%; 3x3: 54.2% ± 1.5%). Based on our conclusion, the difference exists due to the fewer players and thus the larger usable area per player, as found in previous research [5]. In 3x3, attackers have more area to attack the basket or get into a close shooting position during off-ball moves.

There was a significant difference in the effectiveness of long-range shots in the study. Confirming the results of previous studies [3,4], the results also show more long-range-shot attempts in 3x3, but in 5v5, players shoot more effectively from a long range (5v5: 33.8% ± 0.8%; 3x3: 25.2% ± 1.5%). In 5v5, every third long shot is successful, while only every fourth in 3x3, which can be explained in 5v5 by the longer shot clock and thus the tactically more rational selection of the appropriate shooting position. More frequent but less effective shots can still be a rational strategy in 3x3, since the value difference in the two types of shots (close and long-range) is greater than in 5v5.

### 4.3. Shot Selection

Considering the statistical data used in the previous [3,4] and current analysis, it can be clearly stated that the number and efficiency of long-range shots have become a determining factor for either 5v5 or 3x3 basketball games. In today’s men’s basketball teams, it can be observed that almost every team has a centre that regularly experiments with long-range shots [39]. This is even more typical in 3x3 basketball since differences decrease between positions [6] and there are significantly more ball possessions per player in 3x3 [6]. Due to the above, there is a difference between the two disciplines in terms of shot selection too. In 5v5, approximately every third shot is made from a long distance (37.7% ± 0.6%), while in 3x3, it is almost every second shot (44.3% ± 1.2%). Regarding the selection of shots, based on the results of a comparison, we can state that players select the higher-value shot more times in 3x3.

The limitations of this study are that the comparison does not consider different court sizes, shot selections made in different periods of the games or of the shot clock, and the relative position of the defender and of the shooter during shot selection. Therefore, in terms of these, further research is suggested comparing the shots, especially long-range shots, which may explain the differences in shot selection of offensive players between 5v5 and 3x3. In addition, research can be conducted with professional female players so the genders can also be compared.

## 5. Conclusions

Based on the results, we found that changes in game conditions and rules affect player’s thinking, behaviour, and decision-making on the court during the game. In the comparison, in terms of overall shots, we found a significant difference between 5v5 and 3x3 in the shot selection, specifically in the ratio and efficiency of long-range. In 5v5, long-range (three-point) shots have increased in number and effectiveness in basketball games over the decades [39]. Based on the trends, further growth is expected, but longer-term strategies are also expected to change, as long-range shots systematically promise better results for attackers [39] than close-range shots. The results also proved this: in 3x3, the point value of the long-range shot is double compared to the close shot, which motivates the players to shoot more from outside the arc.

Confirming previous results [3,4], the effectiveness of long-range shots is better in 5v5, but players select long-range shots more often in 3x3 compared to overall shots. As a result, it can be stated that the offensive efficiency of 5v5 is better than in 3x3. This is proven by the difference in offensive value. Compared to the maximum points that can be achieved per possession, this value in 5v5 is approximately one point per possession (confirming previous results [38,39]), while in 3x3, this value is one point per two possessions. Using these data, we were able to compare the two disciplines in terms of offensive efficiency, which we named relative offensive value. As a result, we found a significant difference between the relative offensive value, which was 34.7% (± 0.4%) in 5v5 and 27.6% (± 0.8%) in 3x3, which proves that the offensive efficiency is better in 5v5 than in 3x3.

In 3x3, the shorter shot clock and reduction in the number of players require particularly high-speed inertial movement within a limited distance, which results in a relatively high physiological response [5] than in 5v5. In this environment, the players must decide to create a suitable shooting position and to make a shot. Instead of 24 seconds (5v5), 12 seconds (3x3) are available to set up a shooting position, which means that the preparatory movements must be performed faster than in 5v5. Additionally, it also complicates the decision-making situation of the players that the constant change of offensive–defensive roles and unpredictable environment [7] require constant concentration. Therefore, the players’ shot selection, in addition to information processing, is characterized by ecological dynamics and a naturalistic decision-making perspective [11]. Due to the 3x3 games being more intense compared to 5v5, according to the ecological dynamics perspective, the decisions made under higher intensity are made during continuous interactions between the performer and the environment. The behaviour of the players and the team is influenced by the constant contact within the team and adaptation to the opponent [14,15,16,17], which limit decision-making so that the decisions are determined by the structure of the team, the environment, and the player’s abilities. The natural decision-making [18,19] approach is confirmed by time-limited situations (shorter shot clock), which are decisions made under pressure due to constantly changing game situations. The differences between the players’ shot selection and effectiveness suggest that the differences between the game rules of the two basketball disciplines influence the players’ decision-making.

## Figures and Tables

**Figure 1 ijerph-19-15137-f001:**
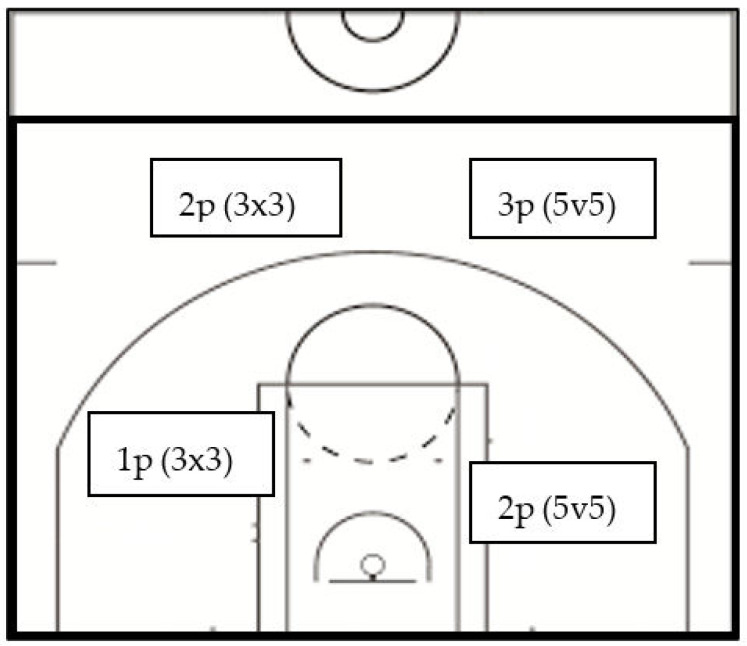
The 5v5 (grey) and 3x3 (black) half-court lines and shooting range of point values for each discipline—adapted from the official FIBA website [7].

**Figure 2 ijerph-19-15137-f002:**
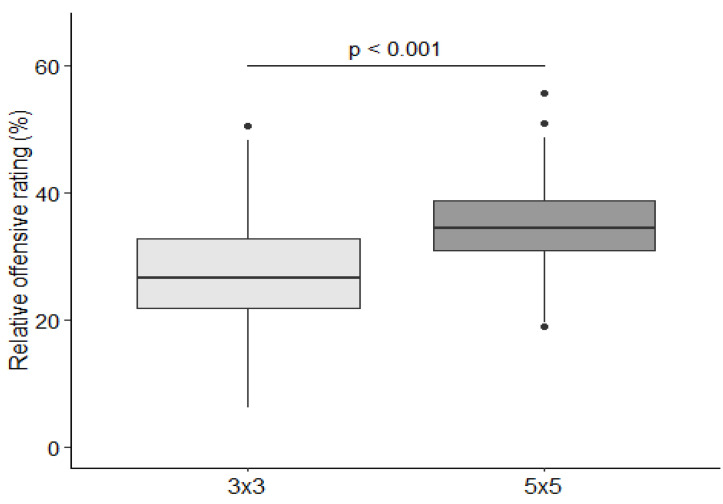
Relative offensive rating regarding basketball discipline.

**Figure 3 ijerph-19-15137-f003:**
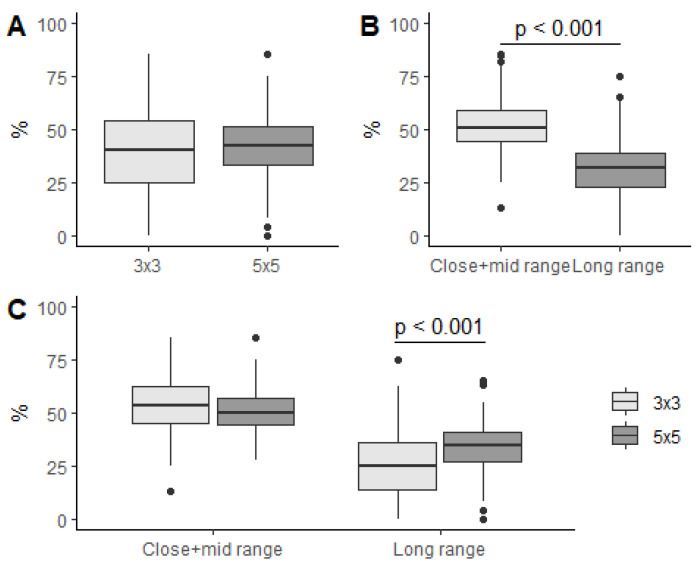
Shooting percentage regarding basketball discipline (**A**), range of shot (**B**), and their interaction (**C**).

**Figure 4 ijerph-19-15137-f004:**
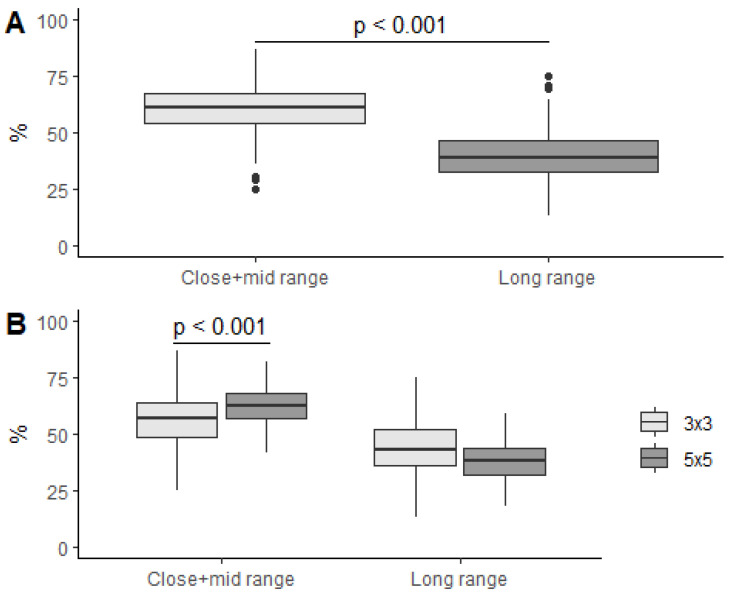
Ratios of close–mid- and long-range-shot attempts to all field-goal attempts overall (**A**) and regarding discipline (**B**).

**Table 1 ijerph-19-15137-t001:** Categorization of games based on score difference [23].

Category	3x3	5v5
Tie	1–4	1–9
Balanced	5–9	10–22
Unbalanced	10–	23–

## Data Availability

Data are available upon reasonable request to the corresponding author.

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
