# Peer review of "A Comparison of 5v5 and 3x3 Men’s Basketball Regarding Shot Selection and Efficiency"

_ijerph, 2022, doi:10.3390/ijerph192215137_

Round 1
Reviewer 1 Report (Previous Reviewer 3)
I believe you have adequately responded to the complete peer review comments.
Author Response
Based on the other reviewer's suggestions, the manuscript was revised, we changed title of the manuscript, I uploaded it as a attachement.

Reviewer 2 Report (Previous Reviewer 1)
the reviewer thank the authors for their efforts. This is an interesting topic which helps trainers increase the performance using the appropriate strategy in basket ball. however some modifications shouls occur in the manuscript.
The introduction section is too long, please consider respect the most important information related to the topic
The conclusion is too long. Put the most important thing which gave the reader a clear idea about his choise when using one of the two strategies.
Line 44 : consider changing «5vg5 » by 5 vs 5
Line 58 : « environments » should be « environment »
Some of the references should be revised and in relation to the text.
Author Response
Response to Reviewer 2 Comments
Point 1: The introduction section is too long, please consider respect the most important information related to the topic
Response: Based on the reviewer's suggestions, the manuscript was modified, and the title was renamed
Point 2: The conclusion is too long. Put the most important thing which gave the reader a clear idea about his choise when using one of the two strategies.
Response: Based on the reviewer's suggestions, the manuscript was modified, and the title was renamed
Point 3: Line 44: consider changing «5vg5 » by 5 vs 5
Response: Based on the reviewer's suggestion, this has been modified in the text
Point 4: Line 58: « environments » should be « environment »
Response: Based on the reviewer's suggestion, this has been modified in the text
Point 5: Some of the references should be revised and in relation to the text.
Response: Based on the reviewer's suggestion, the references have been modified
Based on the reviewer's suggestions, the manuscript was revised, we has changed title of the manuscript, I uploaded the respones to the reviewer comments and the revised manuscript as attachments

Reviewer 3 Report (New Reviewer)
- English should be improved.
- State information about the Institutional reviewing Board approval.
- There are two essential topics that need to be addressed: at first, it is speculated that shooting efficiency reflects decision-making. Nevertheless, passing, among other elements of the individual technical-tactical behavior in the game, is also a decision-making feature. This, it is strongly suggested that the title and the manuscript should be about what the paper is, namely the differences in the effectiveness of the shots and scoring in 3x3 and 5vs5 basketball. Secondly, as mentioned in the Methods (258-260), different group of players (teams) were examined. To satisfy the study design, the study should be conducted on how the same players behave differently when playing 3x3 compared to 5vs5.
- The Introduction is very extended. It is recommended to focus on the theoretical background of the study and to present the necessity to examine the analyzed parameters. The citations of the past research on other sports (but not for rugby) should be kept to minimum or even omitted.
- Present all goals of the study in the proper section of the Introduction (1.4) (for example, a goal is stated in 151-152). Furthermore, present all equations of the game statistics in the Materials and Methods section.
- An hypo9htesis for the goal of the study mentioned in 249-252.
- For better clarity, stress that it is about game statistics and not “statistics”.
- Define the fixed factors for better readership.
- Do the dots in the Figures represent outliers? How where outliers handled in the analysis of the data?
- What do you mean by “subheadings” (345)?
- Mention the limitations of the study and propose recommendations for future research.
- Mention the patents (445).

Author Response
Response to Reviewer 3 Comments
Point 1: English should be improved.
Response: The text has been rewritten according to the English language and style
Point 2: State information about the Institutional reviewing Board approval.
Response: The study was approved by an institutional review board and meets the ethical standards in sports and exercise science research (Harriss, MacSween & Atkinson, 2019).
Harriss, D. J., MacSween, A., & Atkinson, G. (2019). Ethical standards in sport and exercise science research: 2020 update. International Journal of Sports Medicine, 40(13), 813–817. https://doi.org/10.1055/a-1015-3123
Point 3: There are two essential topics that need to be addressed: at first, it is speculated that shooting efficiency reflects decision-making. Nevertheless, passing, among other elements of the individual technical-tactical behavior in the game, is also a decision-making feature. This, it is strongly suggested that the title and the manuscript should be about what the paper is, namely the differences in the effectiveness of the shots and scoring in 3x3 and 5vs5 basketball. Secondly, as mentioned in the Methods (258-260), different group of players (teams) were examined. To satisfy the study design, the study should be conducted on how the same players behave differently when playing 3x3 compared to 5vs5.
Response: It would be worth investigating how the same players behave differently while playing in a 3x3 and a 5vs5 setting. However, this could only be carried out in a „laboratory” setup, that could bias the results. Thus, the findings presented in our study are based on the game statistics of the performance of professional players during 5vs5 and 3x3 World Cups, which gives an overview of the shot selection of players during high-level international games. In our opinion, comparison of the two disciplines in a real competitive environment provides more useful information for basketball experts than, for example, examining the same 10-10 players in 5vs5 and 3x3 artificial game environments.
Point 4: The Introduction is very extended. It is recommended to focus on the theoretical background of the study and to present the necessity to examine the analyzed parameters. The citations of the past research on other sports (but not for rugby) should be kept to minimum or even omitted.
Response: Based on the reviewer's suggestions, the introduction has been modified and citations of the past research on other sports have kept to minimum
Point 5: Present all goals of the study in the proper section of the Introduction (1.4) (for example, a goal is stated in 151-152). Furthermore, present all equations of the game statistics in the Materials and Methods section.
Response: We presented all goals of the study in the Aims and all equations in Materials and Methods
Point 6: An hypohtesis for the goal of the study mentioned in 249-252.
Response: We presented all goals of the study in the Aims
Point 7: For better clarity, stress that it is about game statistics and not “statistics”.
Response: We modified it to "game" statistics in the text
Point 8: Define the fixed factors for better readership
Response: The fixed factors have been defined in the text
Point 9: Do the dots in the Figures represent outliers? How where outliers handled in the analysis of the data?
Response: Statistical outliers were only visually denoted but were not excluded from the analysis as they represent valid values of the sample
Point 10: What do you mean by “subheadings” (345)?
Response: “subheadings” is deleted from the text
Point 11: Mention the limitations of the study and propose recommendations for future research.
Response The limitations of the study and propose recommendations for future research are mentioned in Conclusion
Point 12: Mention the patents (445).
Response: We supplemented the patents
Based on the reviewer's suggestions, the manuscript was revised, we changed title of the manuscript. I uploaded the respones to the reviewer comments and the revised manuscript as attachments

Round 2
Reviewer 3 Report (New Reviewer)
ijerph-1977233-R1
Reviewer comments
Comments on the Author(s) reply to the initial review:
· Point 1: some additional English improvement is needed, see a few examples below:
L12: Both in 5v5…
L13: two basketball disciplines, can
L23: successful in 5v5. Conclusions:
L35: define the 5v5 abbreviation (like 3x3 in L30). Note that in L234 it appears as 5vs5. Please check for an unanimous use of the abbreviation throughout the text.
L38: basics of 3x3 are based
L65-68: please, rewrite this sentence as it is not read well.
L76-77: between performer and environment [13].
L89: Delete “Some”
L97: delete “.” The same in L251.
L191. Rewrite this part of the text, should it be 2 sentences?
L201: choose a different term for “effects”.
L274: (Figure 2).
· Point 2: there are still no ethical statement or related information concerning an IRB approval. See the Journal Instructions for Authors guidelines, section Research and Publication Ethics (https://www.mdpi.com/journal/ijerph/instructions#ethics ).
· Point 8: state the two fixed factors here instead of L271
· Point 11: The limitations of the study are not material for the Conclusions. It is recommended to move them as the last paragraph of the Discussion. Also, the Conclusions are quite extensive. Most of the content is appropriate for the closing of the Discussion prior the limitations of the study. The text of L415-424 should remain, added a brief summery of the basic results and the significance of your findings for coaches and practitioners.
· Point 12: see the Journal guidelines and the article template about the content at this part of the manuscript.
Author Response
Point 1: some additional English improvement is needed, see a few examples below:
L12: Both in 5v5…Both in 5v5 - corrected
L13: two basketball disciplines, can - corrected
L23: successful in 5v5. Conclusions: - corrected
L35: define the 5v5 abbreviation (like 3x3 in L30). Note that in L234 it appears as 5vs5. Please check for a unanimous use of the abbreviation throughout the text. – It is defined in the text
L38: basics of 3x3 are based - corrected
L65-68: please, rewrite this sentence as it is not read well.
It is corrected: In a rugby union study [11], researchers demonstrated that affordances are often perceived during the unfolding action in a game. For example, the movement of a teammate or opponent may can create a new opportunity during the game for another action that overrides the pre-planned action.
L76-77: between performer and environment [13]. - corrected
L89: Delete “Some” - deleted
L97: delete “.” The same in L251. - deleted
L191. Rewrite this part of the text, should it be 2 sentences?
- It is corrected: Comparison to 5v5, there is significantly more ball possession per player in 3x3 [6]. In 5v5, the ball possession rate of the playmaker is the highest. However, in 3x3 the smaller number of players and offensive time (shot clock) reduce the role of the primary ball possessor.
L201: choose a different term for “effects”. - corrected: “influences”
L274: (Figure 2). - corrected
Point 2: there are still no ethical statement or related information concerning an IRB approval. See the Journal Instructions for Authors guidelines, section Research and Publication Ethics (https://www.mdpi.com/journal/ijerph/instructions#ethics).
- in the Patents: The study was approved by the Institutional Research Ethics Committee of Hungarian University of Sports Science (TE-KEB/32/2022)
Point 8: state the two fixed factors here instead of L271
- It is moved to Materials and Methods: Fixed factors were discipline of baseball (two levels: 3x3 and 5vs5) and range of the shots (two levels: close and mid-range shots, long range shots).
Point 11: The limitations of the study are not material for the Conclusions. It is recommended to move them as the last paragraph of the Discussion. Also, the Conclusions are quite extensive. Most of the content is appropriate for the closing of the Discussion prior the limitations of the study. The text of L415-424 should remain, added a brief summery of the basic results and the significance of your findings for coaches and practitioners.
- The limitation of the study was moved to the end of the discussion
Point 12: see the Journal guidelines and the article template about the content at this part of the manuscript.
- Response: We supplemented the patents
This manuscript is a resubmission of an earlier submission. The following is a list of the peer review reports and author responses from that submission.
Round 1
Reviewer 1 Report
This is an interresting comparative study which can be interresting for players and coaches in sports club. The writing is clear but the introduction section seems to be long. Thus i recommed to shorten it.
the comparison of the two field games and the analytical way for the game is interresting for the sport sucess.
page 5, L 196; add the reference please
page 10; L 440: Consider respecting the references form
Reviewer 2 Report
ijerph-1731064
Reviewer comments
In the submitted manuscript, the authors examined the differences in game statistics describing the offense in 3x3 compared to 5v5 basketball. The aim was to use the game statistics as indicators of players’ decision making and thinking.
There are topics that are necessary to be addressed, as pointed below in the General and Specific Comments.
General Comments
- English should be checked throughout the text as minor language issues are present.
- The Introduction is structured in several subsections. Despite the thorough presentation of the factors that are going to be examined, some parts need to be replaced in order to build a solid rationale for the conduction of the study.
- Game statistics could serve as an indirect indicator of players’ decision making and thinking. In order to establish the rationale of the study, further solid research evidence is needed to link players’ decision making and thinking and game statistics, since there are far more factors that determine decision-making within team sports (see for example https://doi.org/10.3390/sports9050065). In addition, as presented by the title of the manuscript and the conclusion stated within the abstract, the purpose of the study should be investigated examining the exact same cohort of players competing in 3x3 and 5v5 basketball. It is recommended that the rationale and the hypothesis of the study should be based on the available data and what the data represent (namely, performance analysis of offense).
- Further elaboration is needed in the Discussion about the role that the defence plays concerning how the opponent acts in offense.
Specific Comments
Title
- See General Comment #3. It is recommended to delete all contents of L2 and to include the term ‘Men’s’ in L4.
Abstract
- L21-23: Once again, see General Comment #3.
Introduction
- See General Comment #2.
- L33-37: This part is suggested to be written after the purpose of the study in subsection 1.4.
- L47-49 This part should be written after L203.
- L65: see General Comment #3. The same for L112.
- L133: Defensive performance indicators should be also included in the analysis. See also General Comment #4.
- L139-144: These data are also repeated in Table 1.
- L165-166: See General Comment #3. The same in L179-181 and L216-218.
- L196: The number of the reference is missing from the squared brackets. It should be 30 or the number of a reference already cited within the manuscript.
Materials and Methods
- L228: use ‘(Table 1)’.
- L230: What do you mean with ‘(*23)’? Please clarify.
- L246: Provide the details for R software (version, manufacturer).
- L254: ‘Level of significance was set at a = 0.05 in all cases’
Discussion
- See General Comment #4.
Conclusions
- L390-392: See General Comment #3.

Reviewer 3 Report
Dear. Author,
Thank you for your very interesting research. I understood that it analyzes same sports with different rules from the stats and reveals the characteristics of the competition. However, I feel that it would be better to check out a few more previous studies for more research development.
Specifically, research on the differences between 7-player and 15-player rugby, research on the characteristics of doubles and singles competitions such as tennis, snowboarding and skateboarding, etc. By including prior research on the performance improvement from experiencing competition in the same sport discipline with different rules, it becomes clear why you are clarifying the characteristics of 5-player and 3-player basketball. Without the above comparative studies, you are merely comparing different sports and stating that there are differences. Why do you conduct this study? Please clarify the significance of the following.
Please add the significance of conducting this study. I will then determine the validity of the research.
Thanks.
Best,
Reviewer 4 Report
Comments and suggestions in PDF.
